# Acute Physiological Responses Following a Bout of Vigorous Exercise in Military Soldiers and First Responders with PTSD: An Exploratory Pilot Study

**DOI:** 10.3390/bs10020059

**Published:** 2020-02-13

**Authors:** Kathryn E Speer, Stuart Semple, Andrew J McKune

**Affiliations:** 1Faculty of Health, Discipline of Sport and Exercise Science, University of Canberra, Canberra 2617 (ACT), Australia; stuart.semple@canberra.edu.au (S.S.); or mckune@ukzn.ac.za (A.J.M.); 2Research Institute for Sport and Exercise Science, University of Canberra, Canberra 2617 (ACT), Australia; 3Discipline of Biokinetics, Exercise and Leisure Sciences, School of Health Sciences, University of KwaZulu-Natal, Durban (KwaZulu-Natal) 4041, South Africa

**Keywords:** post-traumatic stress disorder or PTSD, autonomic nervous system or ANS, heart rate variability or HRV, hypothalamic-pituitary-adrenal (HPA) axis, cortisol, C-reactive protein or CRP, inflammation, chronic disease, exercise

## Abstract

Post-traumatic stress disorder (PTSD) is a prevalent and debilitating condition associated with psychological conditions and chronic diseases that may be underpinned by dysfunction in the autonomic nervous system (ANS), the hypothalamic-pituitary-adrenal (HPA) axis and chronic systemic low-grade inflammation. The objective of this pilot study was to determine psychological, ANS [heart rate variability (HRV)], HPA (salivary cortisol) and inflammatory (salivary C-Reactive Protein) responses to a bout of vigorous exercise in male first responders, military veterans and active duty personnel with (*n* = 4) and without (*n* = 4) PTSD. Participants (50.1 ± 14.8 years) performed a thirteen-minute, vigorous intensity (70%–80% of heart rate max), one-on-one boxing session with a certified coach. Physiological and psychological parameters were measured before, during, immediately after to 30 min post-exercise, and then at 24 h and 48 h post. The effect sizes demonstrated large to very large reductions in HRV that lasted up to 48 h post-exercise in the PTSD group compared with unclear effects in the trauma-exposed control (TEC) group. There were unclear effects for depression, anxiety and stress as well as salivary biomarkers for both groups at all time-points. Findings may reflect stress-induced changes to the ANS for PTSD sufferers.

## 1. Introduction

Post-traumatic stress disorder (PTSD) is a debilitating condition that manifests after experiencing trauma [1]. The lifetime prevalence rate for PTSD amongst Australians that served in the Vietnam War is 20.9%, which is similar to that of United States warzone estimates of 17% [2]. Furthermore, 16.5% of Australian veterans returning from the war in Iraq and Afghanistan are diagnosed with PTSD [2]. Affected individuals suffer distress in their daily lives including functional impairment, negative social consequences and difficulties in maintaining relationships [3]. Additionally, high unemployment rates and medical costs associated with PTSD (approximately $925 million over a two-year period) place a substantial burden on society [3,4,5,6]. Furthermore, chronic diseases related to the gastrointestinal, hepatic, cardiovascular, and respiratory systems as well as sleep disorders, were shown to be significantly higher in Australian Vietnam War veterans with PTSD compared with their non-PTSD, trauma-exposed counterparts [2]. The authors proposed that PTSD should be conceptualized as both a systemic and mental disorder [2]. With a mortality rate that is two to three times greater than that of the general population and physical diseases accounting for the majority of this shortened life expectancy [7,8], understanding the pathology of PTSD is a necessity.

Immune, autonomic nervous system (ANS) and hypothalamic-pituitary-adrenal (HPA) axis mechanisms have been proposed to regulate the association between PTSD and the development of physical co-morbidities such as cardiovascular disease (CVD), gastrointestinal and autoimmune disorders, respiratory, pulmonary, renal, musculoskeletal diseases, and metabolic syndrome [2,9,10,11]. Typically, in the face of immediate danger or stress, activation of the sympathetic nervous system (SNS) within the ANS and increased secretion of catecholamines manifests in the exposed individual both psychologically (e.g., feelings of helplessness, anger, fear) and physiologically (e.g., increased breathing, heart rate, sweating), thereby initiating a negative feedback mechanism [12,13]. Shortly after, the HPA axis mobilizes and results in the release of cortisol from the adrenal gland [14]. Cortisol secretion results in suppression of the immune system, slowing down digestion and inhibiting glucose uptake to direct energy into coping with the immediate stressor [15]. After the stressor has dissipated, circulating cortisol binds to mineralocorticoid (MR) and glucocorticoid receptors (GR), stabilizing the negative feedback mechanism with a return to homeostasis [3,16]. If the stressor is perceived as ongoing, the body is unable to stabilize this hormonal cascade. Consequential development of PTSD may, in part, be due to a buildup of genetic and environmental risk factors. Over time, these risk factors may contribute to gene polymorphisms within the HPA axis, MR and GR, resulting in lower levels of circulating cortisol [3,17]. In this situation, the ANS displays an inappropriate recovery characterized by a delay in the reactivation of the parasympathetic nervous system (PNS) with the SNS remaining activated after the event has ceased [18]. Additionally, due to “cross-talk” between the immune and nervous systems (as suggested by proposed co-evolution and sharing of gene regulation mechanisms), alterations in one system has downstream effects on the other [9]. Specifically, dysregulated ANS activity combined with HPA axis impairment may result in increased systemic inflammation [10]. Considering that a systemic pro-inflammatory state is also present in the pathogenesis of the aforementioned physical comorbidities, it is possible that systemic inflammation may induce physiological wear and tear, driving chronic disease risk in the PTSD population [10].

Given the increased chronic disease risk association within the PTSD population, there are a lack of treatments to manage/prevent chronic disease comorbidities. At present, standard PTSD management revolves around its psychological symptoms, implementing approaches such as cognitive behavioral therapy (CBT), medications [19], and eye movement desensitization and reprocessing (EMDR) therapy [20]. However, these therapies may not be easily accessible and are not without an associated stigma [21]. Considering these limitations, a holistic therapeutic approach encompassing both the psychological and physiological aspects of PTSD may improve treatment outcomes. 

Recent studies on both healthy [22,23] and unhealthy populations have reported positive effects of exercise on mood states, PTSD symptoms (if applicable), as well as physiological and immunological function [21,24,25]. Considering what is known about the alterations in biological mechanisms and systems in those afflicted with PTSD, the anti-inflammatory effects of exercise may help to reduce allostatic load and thereby the comorbidities associated with PTSD [24,26]. Given that heart rate variability (HRV) is an independent and non-invasive marker of PNS/vagal modulation of total ANS activity, examination of HRV reactivity in response to maximal exercise is an important tool for interpreting physical and psychological resilience [27]. Although research into recovery from induced stress (i.e., exercise) and its effects on the overall wellbeing of individuals suffering from PTSD is limited [28], clinical implementation of such a therapy may help to not only manage PTSD symptomatology but also lessen the burden of its incapacitating physical comorbidities [11,19].

This research used a bout of vigorous exercise to help define the acute responses of stress (ANS and HPA axis) and the immune system to a physical stressor in first responder and military veterans and active duty personnel with PTSD and trauma-exposed controls (TEC). The authors hypothesized that the acute negative feedback mechanisms of inflammatory as well as ANS and HPA axis reactivity to the exercise session would be impaired in the PTSD group indicating a delayed or inappropriate response.

## 2. Materials and Methods

This study was approved by The Human Research Ethics Committee, University of Canberra, Research Ethics and Integrity Review Board (project number 17–85). Prior to participation, all participants provided written and signed informed consent.

### 2.1. Study Population

Participants were included if they were male military combat and first responder veterans or active duty soldiers with clinically diagnosed PTSD (n = 4) and TEC (n = 4). The participants with PTSD had been clinically diagnosed by a registered psychologist using the PTSD Checklist. Thus, the investigators had access to participant clinical status but not the clinical interview details such as symptom severity. The ages of participants ranged between 20 to 75 years, inclusive, with no significant differences between groups (Table 1). Participants were recruited through advertising in the local community (newspaper, television and university campus notice boards) and via informational flyers posted in various clinician and hospital waiting rooms, allied health (Vida Health and Rehabilitation) and charity organizations (FearLess Outreach), gyms, as well as other government funded bodies and Australian Defense Force support services (RSL-ACT). Exclusion criteria for the study included clinical diagnosis of renal disease, any immune or endocrine abnormalities, metal implants and musculoskeletal injury or disability restricting the ability to exercise. These exclusion criteria were adhered to throughout the study in order to limit potential confounding variables that may affect systemic inflammation, stress biomarkers and the ability to have a dual-energy x-ray absorptiometry (DEXA) scan and/or participate in vigorous intensity exercise. Participant recruitment and data collection were performed between June and November 2017 and took place at the University of Canberra, Canberra, Australia.

### 2.2. Study Design

The study took place at the University of Canberra Exercise Testing Laboratory (ETL) and Health Hub on four occasions spanning across six days (Figure 1). Participants were told to refrain from exercise at least 24 h prior to the first testing session until completion of all four testing sessions. All testing was performed in the morning between 7:00–11:00 am and each participant was measured within the same hour of their initial testing session. On the initial testing day participants completed two questionnaires including the Exercise and Sports Science Australia (ESSA) adult pre-exercise tool and the Depression, Anxiety and Stress Scale (DASS). Participants also performed a familiarization protocol relating to the physiological and biological sampling (saliva) protocol to be used. Cardiorespiratory fitness was then measured in the ETL through a maximal exercise test. Finally, participants underwent a whole-body DEXA scan situated in the Health Hub. The DEXA scan was incorporated into this study to accurately measure body fat percentage, which is associated with inflammation [30].

Three days later, participants returned to the ETL for the exercise intervention during which they completed a vigorous intensity boxing session. Participants completed the DASS questionnaire immediately before the boxing session and after 30 min of recovery. Physiological testing and saliva sampling were then performed immediately before the boxing session and during the 30 min recovery. Participants returned 24 and 48 h after the boxing session to complete the DASS questionnaires and have physiological measurements and biological samples taken again.

### 2.3. Assessment Methods and Data Acquisition

Participants completed the self-administered ESSA adult pre-exercise screening tool in order to assess their risk of an adverse event during exercise. This tool was also used to record participants’ cigarette smoking status (N non-smokers = 8) [31]. The self-administered DASS questionnaire was used to assess the negative emotional symptoms of depression, anxiety and stress that are associated with PTSD [32,33]. Over 42 items, participants were asked to rate the extent (0–3) to which they were feeling each symptom at the present moment of answering the questions. Scores were then calculated by adding up the numbers correlated with the relevant negative emotional state of depression, anxiety or stress [33]. The DASS questionnaire has been designed to differentiate the three aforementioned negative emotional states and has proven to be useful in managing disorders relating environmental factors and physical or emotional disturbance, such as PTSD [33,34].

All participants completed a pre-participation exercise screening (maximum oxygen consumption–VO_2max_) as an indication of aerobic fitness. The VO_2max_ test was performed on the SRM High Performance Cycling Ergometer (developed in cooperation with the German Cycling Federation, Peter Keen and Dr Wolfgang Stockhausen) with the SRM science PowerMeter, shown to have an accuracy power reading of ±0.5%. Expired air was measured breath-by-breath through indirect calorimetry using a metabolic measurement system (ParvoMedics TrueOne 2400 metabolic cart, Salt Lake City, UT, USA). The gas analyzer was calibrated automatically before every test.

Prior to commencement, participants were given an explanation of the Borg Rate of Perceived Exertion Scale (6–20 RPE scale). The protocol, adapted from Klika et al. [35], started at 50 W and increase by 50 W every two minutes. Participants provided their RPE at the end of each stage (every two minutes). Throughout the test, oxygen intake and carbon dioxide output, ventilation, breathing pattern and respiratory exchange rate (RER) were constantly measured as was heart rate (HR) and blood pressure. Participants were instructed to cycle between cadences of 60–80 rpm and encouraged to reach symptom limited maximal exertion until at least one of the following criteria were achieved: a respiratory exchange ratio (RER) ≥ 1.1, a HR within 10 beats or over their theoretical aged-predicted maximal HR (220-age), an expression of RPE ≥ 16/20 [36]. The maximum HR achieved was recorded for the appropriate prescription and monitoring of the exercise intervention. VO_2max_ was taken as the highest rate of oxygen consumption measured during any 60 s of the test [37]. 

Autonomic reactivity was examined through measuring HRV a total of six times over this experiment. R-R intervals were recorded with the Suunto t6 monitor system using a chest belt that transmits data to a wristwatch (Suunto Inc; Vantaa, Finland). This device has been validated against an ambulatory five-lead electrocardiogram system and the Polar S810i chest belt system in adults [38]. Participants were required to lie supine in the ETL (ambient temperature of 22–25 °C) with legs uncrossed for a 5-min rest period with no external stimulation [39]. Participants’ respiratory rates during the recordings were not controlled for as there is a lack of consensus on the influence of controlled versus non-controlled breathing on HRV parameters, particularly at rates < 10 breaths/min [39]. This is in accordance with the Task Force of the European Society of Cardiology and the North American Society of Pacing and Electrophysiology standards for measurement of short-term HRV [40]. Participants kept the HR belts on throughout the duration of the boxing session. Within the initial five minutes after completion of the boxing session, participants provided a saliva sample in a seated position. Following saliva collection, participants laid supine on a plinth with HRV measurements recorded at 5–10 min, 10–15 min, 15–20 min and 20–25 min post-exercise. At 30-min post-exercise participants performed saliva collection. To improve reliability, R-R intervals were screened for artefacts and were removed and replaced with the mean of the adjacent beats. Due to the effect of IBI data editing on HRV analysis, as described by Sookan and McKune (2012) [41], if missing or ectopic beats exceeded 20% of the values, the recording was not included in the analysis. However, in no cases was there a requirement to exclude the entire recording. Kubios HRV software (Biosignal Analysis and Medical Imaging Group, Joensuu, Finland) was used to analyze linear and non-linear parameters of HRV. The time-domain variables included was root mean square of successive difference of R-R intervals (RMSSD). Low frequency band (LF), high frequency band (HF) and LF/HF ratio were analyzed using both the fast fourier transformation (FFT) and the autoregressive (AR) spectrums. Non-linear data were determined from the Poincare plot and included SD1 and SD2. Acute physiological adjustments to exercise are mediated throughout functional changes in ANS activity and catecholamine production (Åstrand, Rodahl, Dahl, and Strømme, 2003). The spectral analysis (SA) of HRV is commonly considered to be a non-invasive method for the quantification of cardiac ANS related to the sinoatrial node [42]. The HF is entirely modulated by cardiac vagal activity [43,44]. The LF band is associated with baroreflex activity and the bilateral effect of sympathetic and vagal activity on the sinus node [40], and LF/HF ratio reflects the sympathovagal balance [45]. Besides spectral variables, a time domain variable, specifically the square root of the mean of the squares of the successive differences (RMSSD), considered as an index of vagal activity has become very popular among researchers for field HRV analysis [46,47,48,49]. One of the advantages of RMSSD is that it is less sensitive to low breathing frequency than spectral high-frequency power [50]. Salivary cortisol and C-reactive protein (CRP) concentrations were measured to provide information on HPA-axis functioning and inflammation, respectively. CRP was chosen as a salivary biomarker since it is an acute-phase protein associated with inflammation, HRV and chronic diseases such as CVD [51]. Saliva samples were obtained using 2 mL polypropylene cryovials and saliva collection aids (SCA) (Salimetrics, State College, PA, USA). Prior to saliva collection, participants were instructed to refrain from drinking coffee and brushing their teeth the morning of collection and avoid dairy products and sugary/acidic foods 20 min before collection. To further comply with salivary biomarker collection and HRV measurements, participants were not allowed to eat within an hour of their testing session [52]. Saliva samples were obtained between 7:00 am and 11:00 am at baseline, immediately prior to exercise, immediately post-exercise, 30 min, 24 h and 48 h post-exercise. All participants received training in the saliva collection procedure. While seated, participants were requested to lean slightly forward, tilt their heads down and pool saliva in the floor of their mouths for one minute. At the end of the minute, participants swallowed their saliva and were then given the cryovial and SCA to begin passively dribbling saliva through the SCA and into the cryovial. Participants continued in this fashion until they filled the cryovial with 1 mL of saliva and the duration of time for this to be completed was recorded. All samples were then refrigerated at −20 °C until analysis.

Salivary cortisol and CRP were measured using an enzyme-linked immunoabsorbent assay (ELISA) kit (Salimetrics, State College, PA, USA). All samples were analyzed in duplicate. Area under the curve with respect to ground were calculated to estimate the total free salivary cortisol and CRP production during the baseline and post-intervention testing days. Inter- and intra-assay coefficients of variation for cortisol and CRP were < 9%. Participants completed an exercise intervention in the form of a one-on-one 18-min boxing session with a certified boxing coach (Table 2). The boxing program consisted of a 5-min active warm up at moderate intensity [49%–59% of HR max], during which the coach instructed participants on proper stance, footwork, technique and basic punches, and 13 min of boxing at vigorous intensity (70%–80% of HR max) based on results of the cardiovascular fitness testing and monitored using HR (Table 1) [53].

### 2.4. Statistical Analysis

Continuous variables were expressed as means and standard deviations. All HRV parameters were log transformed (Ln) due to their non-normally distributed nature and to allow for parametric statistical analysis [54]. To determine group by time interactions, a two-way analysis of variance (ANOVA) was used for each of the HRV parameters (LnLF, LnHF, LnLF/HF, LnRMSSD) (Figure 2) and for salivary CRP and cortisol (Figure 3). This was to identify any significant differences between the two groups (PTSD vs. TEC) and over the five time-points for the salivary biomarkers (pre-exercise, immediately post-exercise, 30 min, 24 and 48 h), and six time-points for the HRV parameters (pre-exercise, 5–10 min, 10–15 min, 15–20 min, 24 h and 48 h post). Where appropriate, pairwise multiple comparisons were performed using Sidak’s post hoc multiple comparisons test. To examine the variability of the salivary biomarkers, the coefficient of variability (%) was determined at the five time-points for each of the groups. Pearson correlation coefficients were performed to examine the relationships between salivary biomarkers and percent body fat. Statistical analyses were performed using SPSS Version 21 and *p* ≤ 0.05 was considered statistically significant. To facilitate future comparisons between this study and others, the standardized differences were calculated for the between-group comparisons. The effect sizes were interpreted using the Hopkins method of interpreting magnitude: < 0.2 trivial, 0.2–0.6 small, 0.6–1.2 moderate, 1.2–2.0 large, 2.0–4.0 very large, > 4.0 extremely large. If the 95% confidence interval overlapped both small positive and negative effect size values, the magnitude of effect was deemed ‘unclear’ [55].

## 3. Results

### 3.1. Participant Characteristics 

As can be seen in Table 1, all participants were Caucasian, non-smokers and did not report alcohol addiction/abuse. Every participant had a BMI > 25, classifying them as either overweight or obese (BMI ≥ 30). Body fat ranged from 25.7%–40.8%, indicating a high mortality risk [56]. All but one participant demonstrated normal cardiorespiratory fitness for his age. These fitness ranges were derived from “general population” participants across the United States of America [29]. Medication use, co-morbid psychiatric disorders and self-reported physical activity levels were not recorded. Types of trauma included combat, physical assault and a child brutality crime scene. Two participants (1 PTSD, 1 TEC) were still serving at the time of the study. A strong positive correlation was found for CRP and body fat percentage (r = 0.89; *p* = 0.004; very large effect) while body fat percentage was found to be negatively correlated with cortisol (r = −0.29; *p* > 0.05; small effect). 

### 3.2. Exercise Protocol 

All participants completed the boxing session between 70%–80% of their individual max HR with no adverse events.

### 3.3. HRV Analysis

The effects of an acute bout of vigorous intensity boxing on ANS recovery as represented by the parameters for HRV (LnLF, LnHF LnLF/HF, LnRMSSD) over the seven time-points are shown in Table 3. There were no significant group x time interaction effects for any HRV measures (*p* = 0.537, 0.919, 0.926 and 0.771, respectively). 

There was a significant time effect for LnHF with levels decreasing at 5–10 min (*p* = 0.015), 10–15 min (*p* = 0.031) and 15–20 min (*p* = 0.05) post-exercise compared with pre-exercise levels. The decrease in LnHF 20–25 min post-exercise trended towards significance (*p* = 0.061). Effect sizes for LnHF for both groups were similar across all time-points with large decreases in LnHF from pre-exercise through to 25 min post-exercise. Effect sizes for both groups at 24 h and 48 h post-exercise were unclear.

A time effect (*p* = 0.001) was found for LnRMSSD with a significant decrease from pre-exercise to 5–10 min (*p* = 0.005), 10–15 min (*p* = 0.009), 15–20 min (*p* = 0.028) and 20–25 min (*p* = 0.053). Effect sizes for the PTSD and TEC groups demonstrated very large and large decreases, respectively, between pre-exercise and 5–10 min post. Effect sizes for PTSD participants remained very large throughout the 10–15, 15–20, 20–25 min post-exercise with effect sizes unclear at 24 h and a large effect at 48 h post. Effect sizes between pre-exercise and 10–15 min to 48 h post were unclear for the TEC group.

For LnLF there was a borderline significant group (*p* = 0.045) effect with lower levels in the PTSD group and a significant time effect (*p* = 0.006), with LnLF decreasing at 10–15 min (*p* = 0.051) and 20–25 (*p* = 0.046) min post-exercise compared with pre-exercise levels. The decrease in LnLF 15–20 min post-exercise trended towards significance (*p* = 0.064). Based on effect size calculations (Table 3) for pre-exercise versus each of the post-exercise time-points, there was a large decrease in LnLF from pre-exercise to 5–10 min post for both groups. With respect to LnLF for the PTSD group, there were very large decreases until 24 h post-exercise, at which the effect was unclear, with a large decrease again at 48 h post. For the TEC group, effects were unclear between pre-exercise versus all time-points from 10–15 min post through to 48 h post. The effect sizes demonstrate that the boxing session resulted in a decrease in LnLF in the PTSD group that had still not returned to baseline levels by 48 h post while LnLF for the TEC group seemed to recover quickly.

No significant time or group effect was demonstrated for LnLF/HF (*p* = 0.299 and 0.108, respectively). Effect sizes were unclear across all time-points for both groups.

### 3.4. Saliva Analysis

There were no significant group x time interactions (*p* = 0.552), main effects for time (*p* = 0.296) or group (*p* = 0.515) for salivary cortisol. Effect sizes were interpreted as unclear for both groups for pre- versus post-exercise (Table 3). The coefficient of variation (CV%) for salivary cortisol levels across the time-points for the PTSD group ranged from 0.2%–0.5% while the CV% range for the TEC group was 0.5%–0.8%. 

For salivary CRP there was no significant group x time interaction (*p* = 0.368) or main effect for group (*p* = 0.636). However, there was a borderline significant time effect (*p* = 0.051) with an increase from pre-exercise to 30 min post. Effect sizes for both groups across all time-points were interpreted as unclear (Table 3). The CV% range across the time-points was 0.70%–1.2% for the PTSD and 0.9%–1.3% for the TEC group.

### 3.5. Psychological Outcomes 

There were no significant group x time interactions or main effects for time for any of the DASS questionnaire domains (*p* > 0.05). However, there were significant group effects with higher “depression” (*p* = 0.002), “anxiety” (*p* = 0.002) and “stress” (*p* ≤ 0.001) in the PTSD group. Effect sizes examining changes over time for the three DASS domains were unclear for both groups (data not shown).

## 4. Discussion

The primary purpose of this study was to identify the effects of a vigorous intensity boxing session on the acute psychological and physiological recovery process of individuals suffering from PTSD versus TEC. The primary finding was that LnHF and LnRMSSD (markers of vagal activity) had still not recovered by 48 h post-exercise in the PTSD group. Considering that HRV is used as an indicator of autonomic activity [57], the way in which an individual’s post-exercise HRV recovers from exercise is a strong predictor of cardiovascular morbidity and mortality risk [23].

In response to the vigorous boxing intervention for this study, there were large to very large reductions in cardiac vagal activity, as reflected by LnHF and LnRMSSD in both groups, specifically at 5–10 min post-exercise. These findings may indicate that after a vigorous intensity aerobic exercise session there is a delay in cardiac vagal reactivation [54]. In healthy and athletic populations, reactivation of the PNS typically occurs rapidly after exercise termination [23], while individuals with PTSD appear to display a generally decreased PNS tone compared with that of other types of psychiatric diagnoses and healthy populations [22,58,59]. Our study supported this notion, given that there was still a large reduction in LnRMSSD in the PTSD group at 48 h post-exercise, suggesting inadequate recovery. Previous studies investigating the HRV of individuals with PTSD have indicated that reduced vagal activity may reflect altered behavior characteristic of PTSD and SNS tone dominance (e.g., increased anxiety, inability to recognize safe environments) [59,60]. Alternatively, other research has proposed a loss of vagal myelin sheath with chronic stress [61]. One may speculate that the lower cardiac vagal activity in PTSD may affect the efficiency of catecholamine binding to the vagus nerve and the speed at which the signal is then transmitted back to the brain to indicate that the stressful event has ceased, resulting in SNS dominance and a systemically pro-inflammatory state [62]. Although beyond the scope of the current study, future research investigating the relationship between HRV and catecholamine binding to the vagus nerve may improve our understanding of PTSD.

There were large reductions in LnLF in both groups between pre-exercise and 5–10 min post, which continued to decrease in the PTSD group to a very large effect throughout 25 min. While it is uncertain whether the decrease in LnLF was due to baroreflex regulation [63], metaboreceptor input, and/or thermoregulation [54], it may be deduced that this is representative of a decrease in ANS outflow. Additionally, unclear effects of the LnLF/HF response exhibited in both groups post-exercise may be explained by varying fitness levels and/or degrees of psychological stability [45]. Taking into account the unclear effect of exercise on LnLF/HF and the contention that surrounds its depiction of sympathovagal balance [45], more research is required in order to accurately describe the nature of this measure in response to exercise within the PTSD population.

There was no significant difference between groups or “clear” effect of the exercise session on salivary cortisol. Additionally, salivary cortisol was within the normal range for healthy adults for all participants (Salimetrics, State College, PA, USA). However, salivary cortisol readings increased and decreased in a similar pattern in response to exercise for both groups. Considering this was contrary to our hypothesis, it is possible that cortisol reactivity to an acute exercise bout may not be sensitive enough to distinguish between PTSD and TEC [64]. 

Salivary CRP concentrations were not significantly different between groups and effect sizes were unclear. Nonetheless, with fairly analogous pre-exercise levels between groups, the TEC group displayed a gradual increase in CRP from pre to 30 min post-exercise while the PTSD group did not experience much change. This was surprising as previous research consistently describes the PTSD population as displaying a systemically pro-inflammatory state [6,65]. However, with the small sample size in the current study and participants classified as overweight/obese, the clear correlation (*r* = 0.89) between total body fat percentage and salivary CRP may have confounded results [30]. 

The beneficial effects of exercise on mood state and psychological wellbeing are well established [66]. In the present study, the TEC group remained normal and relatively unchanged in all domains of the DASS questionnaire both before and after exercise. There was a trend for the PTSD group DASS domains to decline from pre-exercise throughout 24 h post and then increase back to baseline at 48 h, although effect sizes were unclear. This may suggest that not everyone with PTSD experiences a decrease in their psychological symptoms following exercise. This finding indicates a need for research into different modes, durations and/or intensities of exercise in the PTSD population. Furthermore, it may be beneficial to incorporate another questionnaire to determine if positive mood states improve after exercise rather than a sole focus on negative mood state outcomes.

Compared with the general developed world population, early mortality risk in people suffering from severe mental illness due to physical comorbid disease is substantially greater, cutting approximately 10–17 years off life expectancy [7]. Specifically, CVD seems to be the most commonly associated chronic disease in PTSD [2]. Furthermore, systemic low-grade inflammation may be the underlying link between PTSD and chronic disease [67]. Regarding the current study’s findings, CRP levels did not significantly differ between groups. However, the fact that participants were all classified as overweight/obese, body fat percentage may have confounded results. Despite this, due to the positive correlation demonstrated between body fat percentage and CRP levels in not only this study but in others as well [28,68], inflammation may play a role in chronic disease risk in PTSD sufferers. Given that exercise induces anti-inflammatory effects [26], reduces body fat [69] and decreases the risk of chronic disease acquisition in apparently healthy populations [70], encouraging individuals with PTSD to engage in physical activity may be beneficial.

It is important to note the limitations of this study. Firstly, due to the low statistical power derived from the modest sample size (*n* = 8), our results are limited in their significance. Moreover, the present study’s findings cannot be generalized to women with PTSD or PTSD triggered by traumas other than combat and physical assault. Even though it was beneficial that participants were well matched for age, ethnicity, smoking status, self-reported alcohol intake and cardiorespiratory fitness, the similarities between groups regarding their BMI and body fat percentage may have confounded results. The large age range between participants could have also added bias to the results. Furthermore, we did not control for time since waking prior to testing sessions/biomarker collection, PTSD severity and time since trauma, which have been shown to influence cortisol and pro-inflammatory markers [65,71]. Of note, one participant in the TEC group reported symptoms of PTSD but had no clinical diagnosis. Since clinical diagnosis of PTSD does not involve physiological assessment, possible diagnosis of PTSD may not have been picked up on. Taking this into account, it is important that future research attempts to further define the exact physiological impairments that may constitute PTSD so clinical diagnoses are more accurate.

## 5. Conclusions

Considering the findings presented in the current study, dysregulation in specific physiological mechanisms within the ANS may contribute to the degree of allostatic load and increased chronic disease risk in the PTSD population [72]. This pilot study implemented a unique foundation for comparing the physiological responses to induced stress (i.e., exercise) in first responders and military veterans with PTSD versus TEC, with an indication of reduced cardiac vagal activity demonstrated in the PTSD group. These findings contribute to understanding PTSD as a disorder of not only the mind but the body as well. 

## Figures and Tables

**Figure 1 behavsci-10-00059-f001:**
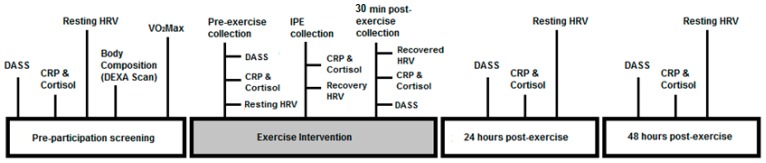
Study Timeline. Note: DASS: Depression, Anxiety and Stress Scale; CRP: C-reactive protein; HRV: heart rate variability, VO_2_max: maximum oxygen consumption: IPE: immediately post exercise.

**Figure 2 behavsci-10-00059-f002:**
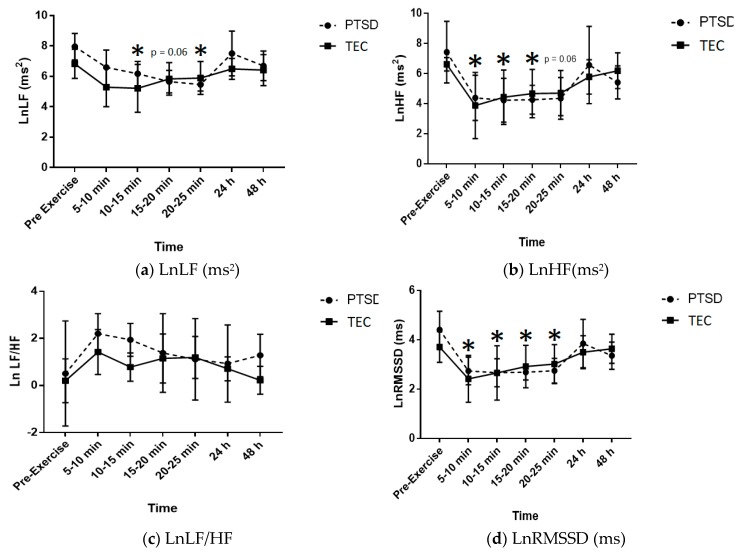
(**a**–**d**): Measured HRV parameter values immediately prior to and following the exercise protocol. **Note:** PTSD: post-traumatic stress disorder; TEC: trauma-exposed controls; LnLF: natural log of low frequency power; LnHF: natural log of high frequency power, LnLF/HF: natural log of low frequency power to high frequency power ratio; LnRMSSD: natural log of the square root of mean squared differences of successive R-R intervals (* denotes significance *p* ≤ 0.05).

**Figure 3 behavsci-10-00059-f003:**
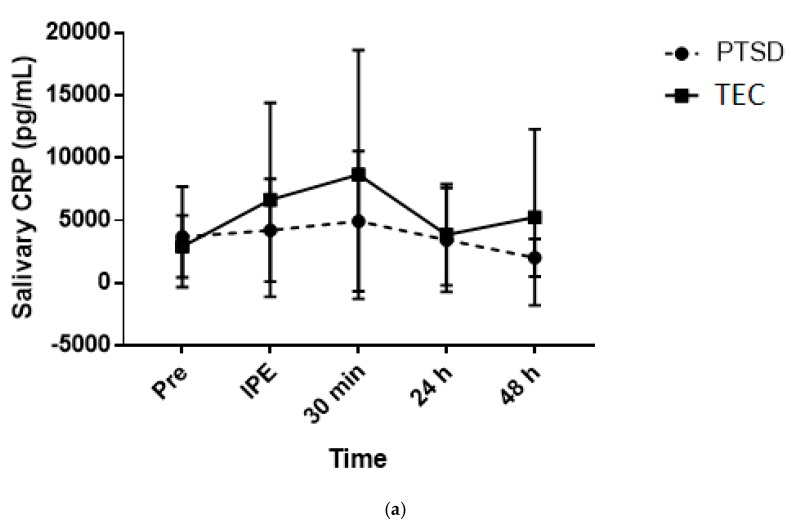
(**a**) Salivary CRP values immediately prior to and following the exercise protocol. Note: PTSD: post-traumatic stress disorder; TEC: trauma-exposed controls; CRP: c-reactive protein; IPE: immediately post exercise. (**b**) Salivary cortisol values immediately prior to and following the exercise protocol. Note: PTSD: post-traumatic stress disorder; TEC: trauma-exposed controls; IPE: immediately post exercise.

**Table 1 behavsci-10-00059-t001:** Participant Characteristics.

Sample Characteristics	PTSD	TEC	Total
	(*n* = 4)	(*n* = 4)	(*n* = 8)
Age (years)	53.30 (13.94)	47 (17.05)	50.13 (14.80)
Mean (SD)			
Ethnicity: Caucasian N (%)	4 (100%)	4 (100%)	8 (100%)
Smoking status: non-smoker N (%)	4 (100%)	4 (100%)	8 (100%)
Self-reported alcoholism N (%)	0 (0%)	0 (0%)	0 (100%)
Type of trauma N (%)			
Combat	3 (37.5%)	3 (37.5%)	6 (75%)
Physical assault	1 (12.5%)	0 (0%)	1 (12.5%)
Child brutality crime scene	0 (0%)	1 (12.5%)	1 (12.5%)
Currently serving N (%)	1 (12.5%)	1 (12.5%)	2 (25%)
BMI (kg/m^2^)	28.43 (1.17)	28.03 (2.91)	28.23 (2.06)
Mean (SD)			
Body fat (%)	29.85 (2.88)	32.05 (6.40)	30.95 (4.74)
Mean (SD)			
VO_2max_ (mL/min/kg) *	35.53 (3.40)	34.50 (8.68)	35.01 (6.13)
HR max (b/min)	167 (14.90)	177 (19.20)	172 (16.78)
(during VO_2max)_			
Mean (SD)			
HR (b/min)	123.25 (10.5)	135.50 (18.45)	129.38 (15.36)
(during boxing session)			
Mean (SD)			

Note: PTSD: Post-traumatic stress disorder; TEC: trauma-exposed controls SD: standard deviation; BMI: body mass index; VO_2max_: maximum oxygen consumption; HR: heart rate. * Reference standards (mean ± SD) for VO_2max_ results (mL/min/kg) based on age range: age 20–29 years, 47.6 ± 11.3; age 30–39 years, 43.0 ± 9.9; age 40–49 years, 38.8 ± 9.6; age 50–59 years, 33.8 ± 9.1; age 60–69 years, 29.4 ± 7.9; age 70–79 years, 25.8 ± 7.1 [29]

**Table 2 behavsci-10-00059-t002:** Boxing Program.

Order	Boxing Combinations	Duration
1	Jab—Cross	1 min
2	Left Hook—Right Hook	1 min
3	Left Upper—Right Upper	1 min
4	Break—standing with deep breathing	1 min
5	Jab—Cross—Left Upper—Right Hook	2 min
6	Break—standing with deep breathing	1 min
7	Jab—Cross—Slip—Cross	2 min
8	Break—standing with deep breathing	1 min
9	Jab—Jab—Cross—Movement	2 min
10	Finish with deep breathing	1 min

**Table 3 behavsci-10-00059-t003:** Effect sizes and interpretations for salivary biomarkers and HRV parameters immediately before exercise versus follow-up measures.

	Time	PTSD	Interpretation	TEC	Interpretation
Cortisol (ug/dL)	Pre vs. IPE	0.20 (−1.20, 1.60)	Unclear	0.10 (−1.30, 1.50)	Unclear
	Pre vs. 30 min	0.60 (−0.90, 1.90)	Unclear	0.10 (−1.30, 1.50)	Unclear
	Pre vs. 24 h	−0.10 (−1.50, 1.30)	Unclear	−0.70 (−2.10, 0.70)	Unclear
	Pre vs. 48 h	−0.10 (−1.30, 1.50)	Unclear	0.40 (−1.0, 1.70)	Unclear
CRP (pg/mL)	Pre vs. IPE	0.13 (−1.26, 1.51)	Unclear	0.65 (−0.77, 2.07)	Unclear
	Pre vs. 30 min	0.26 (−1.14, 1.65)	Unclear	0.80 (−0.64, 2.23)	Unclear
	Pre vs. 24 h	−0.06 (−1.45, 1.33)	Unclear	0.28 (−1.11, 1.67)	Unclear
	Pre vs. 48 h	−0.55 (−1.96, 0.86)	Unclear	0.44 (−0.96, 1.85)	Unclear
LnLF (ms^2^)	Pre vs. 5–10 min	-1.34 (−2.88, 0.19)	Large	−1.34 (−2.87, 0.19)	Large
	Pre vs. 10–15 min	−2.12 (−3.85, −0.39)	Very Large	−1.24 (−2.75, 0.28)	Unclear
	Pre vs. 15–20 min	−2.85 (−4.81, −0.88)	Very Large	−0.97 (−2.44, 0.49)	Unclear
	Pre vs. 20–25 min	−3.63 (−5.88, −1.37)	Very Large	−0.91 (−2.37, 0.55)	Unclear
	Pre vs. 24 h	−0.37 (−1.77, 1.03)	Unclear	−0.39 (−1.79, 1.01)	Unclear
	Pre vs. 48 h	−1.37 (−2.91, 0.17)	Large	−0.41 (−1.81, 0.99)	Unclear
LnHF (ms^2^)	Pre vs. 5–10 min	−1.69 (−3.3, −0.07)	Large	−1.73 (−3.35, −0.10)	Large
	Pre vs. 10–15 min	−1.80 (−3.44, −0.16)	Large	−1.66 (−3.26, −0.05)	Large
	Pre vs. 15–20 min	−1.97 (−3.66, −0.28)	Large	−1.65 (−3.26, −0.05)	Large
	Pre vs. 20–25 min	−1.75 (−3.39, −0.12)	Large	−1.73 (−3.35, −0.10)	Large
	Pre vs. 24 h	−0.36 (−1.76, 1.03)	Unclear	−0.97 (−2.44, 0.50)	Unclear
	Pre vs. 48 h	−1.23 (−2.74, 0.28)	Unclear	−0.48 (−1.89, 0.93)	Unclear
LnLF/HF	Pre vs. 5–10 min	1.0 (−0.47, 2.47)	Unclear	1.30 (−0.23, 2.82)	Unclear
	Pre vs. 10–15 min	0.86 (−0.59, 2.31)	Unclear	0.74 (−0.69, 2.17)	Unclear
	Pre vs. 15–20 min	0.44 (−0.96, 1.84)	Unclear	0.96 (−0.50, 2.43)	Unclear
	Pre vs. 20–25 min	0.30 (−1.09, 1.69)	Unclear	1.09 (−0.40, 2.57)	Unclear
	Pre vs. 24 h	0.21 (−1.18, 1.60)	Unclear	0.68 (−0.75, 2.11)	Unclear
	Pre vs. 48 h	0.45 (−0.95, 1.85)	Unclear	0.03 (−1.36, 1.41)	Unclear
LnRMSSD (ms)	Pre vs. 5–10 min	−2.52 (−4.38, −0.67)	Very Large	−1.61 (−3.20, −0.01)	Large
	Pre vs. 10–15 min	−2.61 (−4.50, −0.73)	Very Large	−1.12 (−2.68, 0.32)	Unclear
	Pre vs. 15–20 min	−3.00 (−5.02, −0.98)	Very Large	−1.05 (−2.53, 0.43)	Unclear
	Pre vs. 20–25 min	−2.60 (−4.49, −0.72)	Very Large	0.97 (−2.44, 0.49)	Unclear
	Pre vs. 24 h	−0.64 (−2.06, 0.78)	Unclear	−0.33 (−1.72, 1.07)	Unclear
	Pre vs. 48 h	−1.60 (−3.19, −0.01)	Large	−0.11 (−1.50, 1.27)	Unclear

Note: PTSD: post-traumatic stress disorder; TEC: trauma-exposed controls; IPE: immediately post exercise; CRP: c-reactive protein; LnLF: natural log of low frequency power; LnHF: natural log of high frequency power, LnLF/HF: natural log of low frequency power to high frequency power ratio; LnRMSSD: natural log of the square root of mean squared differences of successive R-R intervals.

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
