# Peer review of "Acute Physiological Responses Following a Bout of Vigorous Exercise in Military Soldiers and First Responders with PTSD: An Exploratory Pilot Study"

_behavsci, 2020, doi:10.3390/bs10020059_

Round 1

Reviewer 1 Report

Behavioral Sciences (ISSN 2076-328X)

Manuscript ID: behavsci-688262

Type: Article

Title: Acute physiological responses following a bout of vigorous exercise in military soldiers and first responders with PTSD: A pilot study

Comments to the Author

It is thought to be an important and valuable study in the research field related to the occupational group of this study. Manuscript titled “Acute physiological responses following a bout of vigorous exercise in military soldiers and first responders with PTSD: A pilot study” investigated psychological, ANS, HPA and inflammation markers in PTSD after boxing exercise. Participants of in this present study is very interesting and valuable data. However, no effect of boxing session on PTSD participants. I consider that the current manuscript is necessary to revise.

Reviewer’s comments:

Providing criteria for selection participate (the effect size) Materials and Methods section

- Provide sections separately for each content.(eg. 2.1. Participate; 2.2. Study design…)

- Reposition the figure and the table.

Author Response

Reviewer 1: It is thought to be an important and valuable study in the research field related to the occupational group of this study. Manuscript titled “Acute physiological responses following a bout of vigorous exercise in military soldiers and first responders with PTSD: A pilot study” investigated psychological, ANS, HPA and inflammation markers in PTSD after boxing exercise. Participants of in this present study is very interesting and valuable data. However, no effect of boxing session on PTSD participants. I consider that the current manuscript is necessary to revise.

Point 1: Providing criteria for selection participate (the effect size) Materials and Methods section

Response 1: Thank you for this point. We have provided under Section 2.1 “Study population” a clearer description of participant inclusion criteria and how participants were categorized into the PTSD group or the control group. However, since this was a pilot study, a power analysis was not performed.

Point 2: Provide sections separately for each content.(eg. 2.1. Participate; 2.2. Study design…)

Response 2: Thank you for your suggestion. We have now added sub-sections within the methods section.

Point 3: Reposition the figure and the table.

Response 3: Thank you for your comment. We have repositioned the tables and figures so they are centered on the page. We have also enlarged the images for Figures 2a-2d so they are easier for the reader to see.

Reviewer 2 Report

This is an interesting pilot study on the psycho-and physiological responses in patients with PTSD and trauma-exposed controls following exercise.

The paper is well written. However, few points reduced my enthusiasm (mostly methodological).

It is unclear and not specified why the authors chose to study salivary CRP as inflammatory marker. The authors mention in the limitation section that they "did not control for PTSD severity and time since trauma" (line 414) and "one participant in the no PTSD group reported symptoms of PTSD but had no clinical diagnosis" (line 415). This is an important issue. This brings me to ask how PTSD was diagnosed? Did the authors used the CAPS? the PCL? Other? It is difficult to understand how participants reported PTSD symptoms without recording severity of these symptoms or any related information, especially on a study focusing on PTSD! Could the authors details what were the "other" trauma experienced by the participants? It is not clear why the participants went through DEXA scan? It is neither justified anywhere why it would be important (eg no mention of bone densitometry in the introduction), and no results related to the scans seem to be presented. Most of the HRV results presented barely reach significance (p=0.05; although the authors mentioned they set the level of significance at p≤0.05...).  Overall, due to the (very) limited sample size and the marginal significance of their results (although the authors provide effect size for their results, which is probably more meaningful than a p-value), the authors should be more careful on the strength and of their results and on their claims in the discussion; not just in the limitations section.

Minor points: as all subjects were exposed to trauma, it would be easier for the reader if the authors referred to their control group as a trauma-exposed control (TEC) group.

Author Response

Point 1: It is unclear and not specified why the authors chose to study salivary CRP as inflammatory marker.

Response 1: Thank you for this pointing this out. We amended this by providing an explanation (Line 257) for why we chose to study salivary CRP.

Point 2: The authors mention in the limitation section that they "did not control for PTSD severity and time since trauma" (line 414) and "one participant in the no PTSD group reported symptoms of PTSD but had no clinical diagnosis" (line 415). This is an important issue. This brings me to ask how PTSD was diagnosed? Did the authors used the CAPS? the PCL? Other? It is difficult to understand how participants reported PTSD symptoms without recording severity of these symptoms or any related information, especially on a study focusing on PTSD!

Response 2: Thank you for your comment. Since none of the investigators for this study were registered psychologists, we felt it would be more reliable if we only recruited the PTSD participants if they had been clinically diagnosed with PTSD by a registered psychologist. On the other hand, while the control participants experienced trauma, they either did not receive a clinical diagnosis of PTSD by a registered psychologist or have never experienced/reported post-traumatic stress symptoms. We have now specified on Line 111 that the clinical diagnosis was performed by a registered psychologist, which included the use of the PCL. Due to the requirement for PTSD participants to have a clinical diagnosis by a registered psychologist, we did not perform any clinical diagnoses ourselves nor did we have access to participant PTSD symptom severity, etc.

Point 3: Could the authors details what were the "other" trauma experienced by the participants?

Response 3: Thank you for your question. The “other” trauma was experienced by one the control participants. This participant was a police officer. The trauma he experienced was related to witnessing a specific child-brutality crime scene after the crime took place. To his knowledge, the crime scene did not affect him until years later when he became a father. To make this clearer for the reader, we have replaced “other” for “child brutality crime scene” in Table 1. We have also explicitly included it in our description for the types of trauma experienced by participants in Section 3.1, Line 338.

Point 4: It is not clear why the participants went through DEXA scan? It is neither justified anywhere why it would be important (eg no mention of bone densitometry in the introduction), and no results related to the scans seem to be presented.

Response 4: Thank you for your point. The DEXA scan was used to accurately measure body fat percentage, as opposed to bone density, which has been linked with inflammation. We have revised the manuscript (Line 148) to reflect this with referencing. 

Point 5: Most of the HRV results presented barely reach significance (p=0.05; although the authors mentioned they set the level of significance at p≤0.05...).  Overall, due to the (very) limited sample size and the marginal significance of their results (although the authors provide effect size for their results, which is probably more meaningful than a p-value), the authors should be more careful on the strength and of their results and on their claims in the discussion; not just in the limitations section.

Response 5: Thank you for bringing this to our attention. We agree with you and, as such, have tempered our claims/language throughout the manuscript (especially the results, discussion and conclusions).

Minor points:

Point 6: as all subjects were exposed to trauma, it would be easier for the reader if the authors referred to their control group as a trauma-exposed control (TEC) group.

Response 6:  Thank you for your point. We have replaced the original “No PTSD” with your suggested terminology “trauma-exposed controls (TEC)” throughout the abstract and manuscript.

Reviewer 3 Report

The following study investigated acute physiological and psychological responses to vigorous exercise in PTSD individuals. Overall, the sample size is small, but I like the focus of the study. The authors are commended for the completion of an overall excellently written manuscript. Despite the sample size, I think there is still possible conclusions to be made. However, I do have some concerns regarding methods and analysis. Below is my comments:

Line 39: Sentence is long and hard to read as written. Split up into two sentences.

Line 55: Use commas instead of “and” repeatedly

Line 57: I think I understand what the authors are trying to say here but authors should be more specific about the “glucose metabolism” portion as it inhibits glucose uptake and promotes free fatty acid oxidation rather than glucose breakdown.

Line 74: target what exactly? The disease risk?

Line 77: I think authors should add a conclusive statement here to help with flow. For example, a pragmatic therapeutic strategy targeting psychological and physiological aspects of PTSD is needed….which leads nicely into the exercise portion.

Line 89-91: Authors seem to be basing this hypothesis based on the gene polymorphisms mentioned earlier. But if this is an acute response would this not be more concerned with negative feedback mechanisms? The authors are not incorrect, but clarification on negative feedback mechanisms may be warranted.

Overall, the introduction is clear, concise, and informative.

Participants and measurements need to be described more clearly. Below are my concerns:

How often did the participants participate in exercise? Were participants told to not exercise prior to HRV measurements? Were participants told to not drink caffeine or any stimulants prior to measurement? Authors state that testing occurring between 7 am- 11 am but did each participant complete their trials at the same time? i.e. is it possible that someone did their first trial at 7 am and their second at 11 am? Was HRV measured immediately upon waking? To my knowledge, true HRV values are obtained immediately upon waking.

Line 162: Authors should report percentiles for age group to give the reader a better insight to training status

Line 174: Is there a reason why authors chose to use cycling when data collection is occurring on full body exercise?

Line 220: Is there justification of letting participants eat before their HRV measurements?

Line 261: The font and symbols are incredibly small making it hard to interpret.

Line 326-342: I think additional analysis may be needed. The standard deviations appear to be high even at baseline. Thus, analyzing the absolute change (baseline-experimental value) for the time points may be more appropriate. Or, normalize all values to baseline and report fold changes. Either way, it is difficult to tell if there is really not a difference with such large variance. Obviously sample size is most likely an issue here, but authors should see what happens when taking into account a variable baseline.

Line 359: I like this rationale, but this is pure speculation and cannot be answered based on your data. It is called a discussion for a reason but just preface to the reader that you didn’t measure it and suggest future research is needed. Also, authors should try to find explanations that their data can actually support especially since this is the main finding.

Line 373: what does “normal” mean? Normal for trained individuals? Normal for vigorous intensity exercise?

Line 375: Did authors group everyone together and do this analysis? This should be provided. Also, this is counter to your hypothesis in that you provided rationale for impaired HPA responses. Thus, you should address this in the discussion.

Line 382: But what about training status? If they train regularly should that not in part negate this?

Author Response

Point 1: Line 39: Sentence is long and hard to read as written. Split up into two sentences.

Response 1: Thank you for your suggestion. We agree and have rewritten and condensed this sentence (Line 40).

Point 2: Line 55: Use commas instead of “and” repeatedly

Response 2: Thank you for bringing this to our attention. We have replaced “and” with commas to improve the flow of the sentence for readers (Line 58-59).

Point 3: Line 57: I think I understand what the authors are trying to say here but authors should be more specific about the “glucose metabolism” portion as it inhibits glucose uptake and promotes free fatty acid oxidation rather than glucose breakdown.

Response 3: Thank you for your suggestion. We agree that this could be confusing and have now edited the sentence to state, “inhibiting glucose uptake” rather than “increasing glucose metabolism” (Line 62).

Point 4: Line 74: target what exactly? The disease risk?

Response 4: Thank you for your point. We have clarified (line 79) that there are a lack of treatments for managing the chronic disease risk.

Point 5: Line 77: I think authors should add a conclusive statement here to help with flow. For example, a pragmatic therapeutic strategy targeting psychological and physiological aspects of PTSD is needed….which leads nicely into the exercise portion.

Response 5: Thank you for this comment. We have added in a conclusive statement beginning on Line 83.

Point 6: Line 89-91: Authors seem to be basing this hypothesis based on the gene polymorphisms mentioned earlier. But if this is an acute response would this not be more concerned with negative feedback mechanisms? The authors are not incorrect, but clarification on negative feedback mechanisms may be warranted.

Response 6: Thank you for your comment. We have edited the introduction of the manuscript to differentiate between gene polymorphisms and negative feedback mechanisms (Page 2, 2nd paragraph) and revised our hypothesis to mention that this hypothesis would be investigating the acute negative feedback response to stress (Line 100).

Point 7: Overall, the introduction is clear, concise, and informative.

Response 7: Many thanks for your compliment.

Participants and measurements need to be described more clearly. Below are my concerns:

Point 8: How often did the participants participate in exercise? Were participants told to not exercise prior to HRV measurements? Were participants told to not drink caffeine or any stimulants prior to measurement? Authors state that testing occurring between 7 am- 11 am but did each participant complete their trials at the same time? i.e. is it possible that someone did their first trial at 7 am and their second at 11 am? Was HRV measured immediately upon waking? To my knowledge, true HRV values are obtained immediately upon waking.

Response 8: Thank you for these questions. The levels of physical activity were not recorded for participants. However, they were instructed to not exercise prior to HRV measurements (i.e. 24 hours before the VO2max and for the duration of the intervention), which we have now described on Line 140. Notably, none of the participants were exercising around the time of the study (i.e. in the weeks leading up to the study). They were also told not to eat or drink caffeine/any stimulants prior to each testing session (Line 259). Additionally, participants completed each session within the same hour of their initial testing session (added on Line 142). With respect to true HRV values being obtained upon waking, we agree with your statement and we have now added the fact that we were unable to collect HRV upon waking as a limitation (Line 484).

Point 9: Line 162: Authors should report percentiles for age group to give the reader a better insight to training status

Response 9: Thank you for this suggestion. We have referenced the research conducted by Kaminsky et al. 2015 “Reference standards for cardiorespiratory fitness measured with cardiopulmonary exercise testing: Data from the fitness registry and the importance of exercise national database” to report the mean ± SD for VO2max based on age range (please see the note below Table 1). We also added a statement in “Section 3.1 Participant characteristics” to categorize participant cardiorespiratory fitness based on age range.

Point 10: Line 174: Is there a reason why authors chose to use cycling when data collection is occurring on full body exercise?

Response 10: Thank you for your question. We were advised by our Human Ethics Committee that since some of our participants were older (some were in their 70’s) and classified as “vulnerable” that they recommended performing the VO2max on a bicycle rather than a treadmill for safety reasons.

Point 11: Line 220: Is there justification of letting participants eat before their HRV measurements?

Response 11: Thank you for this comment. The reason why we allowed participants to eat at least an hour before their testing session was an ethical requirement since some participants were being tested later in the morning (e.g. 10:00am).  This was specifically to prevent food intake impacting the salivary biomarkers. We have now revised Line 262 to specify this with an additional reference to support it.

Point 12: Line 261: The font and symbols are incredibly small making it hard to interpret.

Response 12: Thank you for bringing this to our attention. We have enlarged the images for Figures 2a-2d so they are easier for the reader to see.

Point 13: Line 326-342: I think additional analysis may be needed. The standard deviations appear to be high even at baseline. Thus, analyzing the absolute change (baseline-experimental value) for the time points may be more appropriate. Or, normalize all values to baseline and report fold changes. Either way, it is difficult to tell if there is really not a difference with such large variance. Obviously sample size is most likely an issue here, but authors should see what happens when taking into account a variable baseline.

Response 13: Thank you for this suggestion. We, too, were concerned with our small sample size and wide ranges of standard deviations. Given these factors, we performed effect size calculations (please refer to Table 3) to provide additional analysis for the small sample size.

Point 14: Line 359: I like this rationale, but this is pure speculation and cannot be answered based on your data. It is called a discussion for a reason but just preface to the reader that you didn’t measure it and suggest future research is needed. Also, authors should try to find explanations that their data can actually support especially since this is the main finding.

Response 14: Thank you for pointing this out. We have now included similar research to help discuss and support our main findings of reduced HRV in the PTSD group (line 419). For the speculative theory, we have added in a statement mentioning that the hypothesis is beyond the scope of our study and have suggested that future research is needed for this explanation (Line 427).

Point 15: Line 373: what does “normal” mean? Normal for trained individuals? Normal for vigorous intensity exercise?

Response 15: Thank you for your question. The “normal” range was based off of the company Salimetrics which provides the saliva assay kits and cortisol ranges for healthy adults. We have now specified for the reader that all participants fell within the normal (i.e. healthy) range for adults (Line 440).

Point 16: Line 375: Did authors group everyone together and do this analysis? This should be provided. Also, this is counter to your hypothesis in that you provided rationale for impaired HPA responses. Thus, you should address this in the discussion.

Response 16: Thank you for this comment. Yes, we grouped both the PTSD and trauma-exposed controls together for both the salivary cortisol and CRP analysis, which we described in Section 3.4 “main effects for time”. We have now included a statement describing that our salivary cortisol results were contrary to our hypothesis and provided a possible reason for this.

Point 17: Line 382: But what about training status? If they train regularly should that not in part negate this?

Response 17: Thank you for your question. We obtained a measure of training status via VO2max testing to give an indication of participant fitness levels. We have now added in these explanations as mentioned in our response to Point 9. However, training status in terms frequency/duration/intensity was not recorded as data for this study. Although most participants would have been classified as “sedentary” around the time of the study since they did not participate in physical activity or had not been exercising in the month or two leading up to the study. Since the cardiorespiratory fitness levels of the participants were mostly within range (except for one participant) and none of the participants were exercising around the time of the study, we feel training status would not have negated our results.  

Round 2

Reviewer 1 Report

This time, the revision file is confirmed to have responded well to the reviewer's request. 

Author Response

Point 1: This time, the revision file is confirmed to have responded well to the reviewer's request

Response 1: Thank you for your time in reviewing and approving our revised manuscript.

Reviewer 2 Report

Thanks for revising the manuscript. The authors addressed my concerns.

There are just few minor points:

Table 3 is not presented in the downloaded version of the manuscript I reviewed (line 362). It would be great if the authors use consistent decimals through the manuscript when reporting their statistics. It would be better here to use 3 decimals. Clarify in the Methods section that the authors did not have access to the details of the clinical interview (for symptom severity), and that they only considered the clinical status according to the interview (lines 102-4) Maybe clarify in the title and in the manuscript that the study is exploratory (in addition to be a pilot study)

Author Response

Reviewer 2:

Point 1: Table 3 is not presented in the downloaded version of the manuscript I reviewed (line 362).

Response 1: Our apologies that you did not receive Table 3 in your version of the manuscript. It is provided in the manuscript, however, if you are having trouble viewing it we have also uploaded it (entitled "Table 3") as a word doc attachment in this response.

Point 2: It would be great if the authors use consistent decimals through the manuscript when reporting their statistics. It would be better here to use 3 decimals.

Response 2: Thank you for this comment. We have amended the reporting of our statistics throughout the results section to 3 decimal places.

Point 3: Clarify in the Methods section that the authors did not have access to the details of the clinical interview (for symptom severity), and that they only considered the clinical status according to the interview (lines 102-4)

Response 3: Thank you for your point. We have added that we only had access to clinical status and not clinical interview details to Line 112.

Point 4: Maybe clarify in the title and in the manuscript that the study is exploratory (in addition to be a pilot study) 

Response 4: Thank you for your suggestion. We have added to the title that this is an exploratory study.

Reviewer 3 Report

The authors have done a good job of addressing my concerns. Good work.

Author Response

Point 1: The authors have done a good job of addressing my concerns. Good work.

Response 1: Thank you for your time in reviewing and approving our revised manuscript.